# Association of Scapular Dyskinesis with Neck and Shoulder Function and Training Period in Brazilian Ju-Jitsu Athletes

**DOI:** 10.3390/medicina59081481

**Published:** 2023-08-17

**Authors:** Ji Hyeon Jeong, Young Kyun Kim

**Affiliations:** Graduate School of Sports Medicine, CHA University, Seongnam 13496, Republic of Korea; jeong8972@naver.com

**Keywords:** Brazilian ju-jitsu (BJJ), scapular dyskinesis (SD), shoulder strength, neck pain

## Abstract

*Background and Objectives*: Neck and shoulder injuries are common in Brazilian ju-jitsu (BJJ) athletes, and scapular dyskinesis (SD) is associated with these injuries. This study aimed to investigate the prevalence of SD in BJJ athletes, their neck and shoulder function and strength, and the BJJ training period. *Materials and Methods*: Forty-eight BJJ athletes participated in the study. Years of experience with BJJ, belt, shoulder internal and external rotation strength, neck strength, neck disability index (NDI), and SD were measured. *Results*: Approximately 31 BJJ athletes (64.6%) showed SD, and the nondominant arm showed a more obvious SD (*n* = 22, 45.8%) than the dominant arm (*n* = 18, 37.5%). Those with over five years of BJJ training experience showed a significantly higher rate of SD (*p* = 0.006) than those with less than five years of experience. Shoulder isometric internal rotation strength was significantly weaker in the obvious SD group than in the normal SD group (*p* = 0.014). Neck isometric strength and NDI did not differ significantly between individuals with or without SD. *Conclusions*: SD was common among BJJ athletes, and more experienced BJJ athletes exhibited higher rates of SD. Shoulder rotational strength was weaker with SD. Further studies are necessary on the neck and shoulders of BJJ athletes with SD.

## 1. Introduction

Brazilian ju-jitsu (BJJ) is a grappling martial art emphasizing takedowns, joint locks, and chokeholds to win with different fight styles [1,2]. Although BJJ athletes are not exposed to direct blows and kicks, the prevalence of injuries such as fractures, dislocations, sprains, muscle strains, and tendinopathy is very high [3,4,5]. According to Nicolini et al. (2021), 84% of BJJ athletes experience orthopedic injuries in two years, with an average of 58 days to recover from injuries [5]. Non-strike combat sports such as judo and wrestling showed higher neck (28.4%) and shoulder (45.6%) injury rates than strike combat sports such as boxing and taekwondo (neck = 14.7%, shoulder = 12.3%) [6,7,8]. Faira et al. (2022) [4] reported a shoulder injury rate of 16.1% to 49.2% among BJJ athletes. In addition, neck (50.8%) and shoulder injuries (49.2%) were very common with BJJ training [1]. Seventy-five percent of shoulder injuries in BJJ athletes occur during sweeping and tumbling [9]. However, sweeping and tumbling can lead to scoring and winning matches [10,11]. Therefore, BJJ athletes are willing to use those techniques, and they are at risk of sustaining shoulder injuries [9]. Rotator cuff injury and acromioclavicular joint separation are the most common shoulder injuries in BJJ athletes [11]. Seventy-three percent of acromioclavicular (AC) joint separations showed scapular dyskinesis (SD), which is a malfunction of the scapula [12]. SD causes neck pain and disability [13]. A high rate of neck and shoulder injuries in BJJ athletes might be related to SD, which is associated with neck pain and shoulder injuries.

SD is defined as abnormal scapular position and/or scapular motion [14]. SD can reduce shoulder function efficiency and interact with shoulder pathologies [15]. The presence of SD increases the risk of shoulder pain by 43% over two years [16]. Overhead athletes showed a higher prevalence rate (54.5%) of SD than non-overhead athletes (33.3%) [17]. However, non-overhead athletes were limited to a few sports and combat sports were not included [17]. One SD prevalence study investigated 30 male athletes, including judo and BJJ athletes; however, handball, volleyball, and swimming athletes were also included [18]. Therefore, the prevalence of SD in BJJ athletes remains unclear. The prevalence of SD during boxing was 52.7% (*n* = 38/72) [13]. Boxing is a strike combat sport, whereas BJJ is a grappling combat sport. Therefore, the prevalence of SD and related disabilities may differ.

The scapula is a bridge connecting the neck and shoulders that provides mobility and stability [19]. The cervical spine and the scapula are related; therefore, problems in any of these regions can affect each other [20]. The scapulothoracic muscles transfer loads between the cervical spine and upper limbs, and neck pain can alter the scapulothoracic muscles, causing stiffness and SD [20,21]. Neck disability worsened significantly as the severity of SD increased in elite boxers [13]. Neck pain alters scapular motion during arm elevation [22]. BJJ has a higher neck injury rate than judo or kickboxing [11,23]. Therefore, BJJ athletes are more exposed to SD; however, research on this topic is lacking.

Neck and shoulder injuries are the most common in BJJ athletes [1,24,25]. SD has been reported in many shoulder injuries such as AC joint sprain and impingement [16,26]. The lack of research on SD in BJJ athletes could lead to misleading treatment and rehabilitation outcomes due to neck and shoulder injuries. Therefore, this study aimed to investigate SD in BJJ athletes with neck and shoulder pain and disability and the relationship of SD with strength and duration of BJJ training. Our hypothesis was that BJJ athletes with SD would present more neck and shoulder pain and disability with weaker neck and shoulder strength.

## 2. Materials and Methods

### 2.1. Participants and Study Design

This was a cross-sectional, single-blind study. We recruited 48 BJJ athletes from 5 teams in Seoul and Kyeonggi, Republic of Korea. The inclusion criteria were more than 1-year BJJ-trained males or female athletes between 20 and 50 years of age, with at least one official BJJ competition participation. The exclusion criteria were acute injury within 1 month or a history of upper body orthopedic surgery. The sample size was calculated using G Power software version 3.1 (University of Kiel, Kiel, Germany). With an effect size of 0.8 from the previous study [27], a significance level of 0.05, and a power of 0.95, we required a sample size of 30. Fifty-one BJJ athletes were recruited for this study, and three were excluded because of acute injuries at the time of measurement. Written informed consent was obtained from all participants prior to measurements. The Ethical review board of CHA University approved this study (1044308-202209-HR-048-02).

### 2.2. Protocol

The age, height, weight, years of experience with BJJ, and belt of all participants were obtained. The scapular dyskinesis test (SDT) was used to evaluate SD. Internal and external shoulder rotation strength, neck flexion extension, and lateral flexion strength were measured. The neck disability index (NDI) questionnaire was used to assess neck disabilities.

### 2.3. SDT

We followed the McClure [28] SDT method with the palpation test described by Huang [29]. The participants were asked to wear a T-shirt on the top to be tested. Those under 68 kg were asked to hold 1.5 kg dumbbells in both hands, and those over 68 kg used 2 kg dumbbells. While standing straight, the participants were asked to flex their shoulders to 180° for 3 s and then extend their shoulders to 0° for 3 s. The clinician palpated the scapula behind the participant to evaluate SD as normal (normal scapular movement), subtle (subtle abnormal movement), or obvious (obvious abnormal movement) [28,29]. The researcher who measured SD had 15 years of experience in shoulder research and rehabilitation and was blinded to other measurements.

### 2.4. Shoulder Rotation Isometric Strength

A ForceFrame with a fixed upper-limb mold (VALD, Brisbane, Australia) was used to measure the shoulder internal and external rotation isometric strengths. The participant was placed in the supine position and the tested shoulder was abducted at 90° with the elbow flexed at 90°. The participant was asked to rotate the shoulder internally and externally for two-time practice with 50% strength. They were then asked to rotate with full strength (Figure 1) internally and externally. We measured each rotation two times with a 10 s break between measurements. The average of two measurements was recorded for data analysis. The test reliability measurements were reported as high (ICC = 0.85–0.95) [30].

### 2.5. Neck Isometric Strength

ForceFrame (VALS, Brisbane, Australia) was used to measure the isometric neck strength. The participants were asked to be in a quadruped position with their head on ForceFrame. The frames were adjusted for each participant. Flexion, extension, and left and right lateral flexions were measured (Figure 2). After two-time practice sessions at 50% strength, two full isometric strength measurements were performed with a 10 s break between measurements. The average of the two measurements for each move was recorded. The reliability of this method was reported to be high (ICC = 0.92–0.97) [31].

### 2.6. Neck Disability Measure

Participants were asked to complete the NDI to measure their neck disabilities. A higher NDI score indicates higher neck disability, with the total score being between 0 and 50 [32]. The reliability of the test was high (ICC = 0.89) [33], and we used the Korean version which also had a high reliability (ICC = 0.927) [32].

### 2.7. Statistical Analysis

The one-way ANOVA was used to compare the average neck disability, neck and shoulder strength, and BJJ training period according to SD severity. The Shapiro–Wilk test was used to test the normal distribution of the data. The Mann–Whitney U test was used for non-normally distributed data. The Bonferroni post hoc test was used to compare each group when the data were significantly different. The level of significance was set at a *p*-value of less than 0.05. IBM SPSS 26.0 version (SPSS Inc., Chicago, IL, USA) was used for all statistical analyses.

## 3. Results

### 3.1. The Prevalence of Scapular Dyskinesis in BJJ

A total of 48 BJJ athletes participated in the study. Thirty-one BJJ athletes (64.6%) showed SD in at least one shoulder (normal, 19.8%; subtle, 38.5%; obvious, 41.7%). The non-dominant arm showed a more obvious SD (*n* = 22, 45.8%) than the dominant arm (*n* = 18, 37.5%) (Table 1).

### 3.2. Association with BJJ Training Period and Belts

The longer the BJJ athletes trained, the higher the SD rates in the non-dominant arm (*p* = 0.006) as well as the color of the belts (*p* = 0.015) (Table 2 and Table 3). However, no significant difference was found in the dominant arm (training period *p* = 0.742, belt color, *p* = 0.550).

### 3.3. Shoulder Rotation Isometric Strength

There were significant differences with shoulder isometric internal rotation strength among normal, subtle, and obvious SD in the dominant arm (normal = 100.43 N ± 24.23, subtle = 96.43 ± 20.95, obvious = 77.83 ± 21.80; *p* = 0.014; normal > obvious, subtle > obvious) (Table 4). The dominant arm external rotation isometric strength was not significantly different (*p* = 0.054) but showed a trend that the normal SD group was stronger, and the obvious SD group was weaker (Table 4). The isometric strengths of internal and external rotations of the other shoulders were not significantly different.

### 3.4. Neck Isometric Strength

There were no significant differences in neck isometric flexion, extension, and lateral flexion between the dominant and non-dominant arm SD groups (Table 5).

### 3.5. NDI

Although NDI scores increased as the severity of SD increased in the dominant arm (normal = 3.83 ± 4.6, subtle = 4.72 ± 4.3, obvious = 5.44 ± 4.6), there was no significant difference between the dominant arm SD (*p* = 0.515) or non-dominant arm SD (*p* = 0.494) (normal = 5.71 ± 5.9, subtle = 3.53 ± 2.6, obvious = 5.55 ± 5.1).

## 4. Discussion

This study aimed to investigate the prevalence of SD in BJJ athletes according to neck disabilities, isometric strength, shoulder rotational isometric strength, and training period. We found that 64.6% (31/48) of the BJJ athletes had SD. The prevalence of obvious SD was higher in the non-dominant arm (*n* = 22, 45.8%) than in the dominant arm (*n* = 18, 37.5%). According to a previous study, the prevalence of SD in overhead athletes was 61% [17]. Therefore, the prevalence of SD in BJJ athletes was as high as that in the overhead athletes even though BJJ is a non-overhead sport. Compared with elite boxing (52.7%), BJJ athletes had a higher prevalence of SD [13]. The high shoulder injury rate is due to sweep, submission, and tumbling [3,9,25]. Rotator cuff injuries and AC sprains are common [2,11] in BJJ, and SD is frequently observed in these injuries [12,15,34]. This may be the reason for the high prevalence of SD in BJJ athletes. However, there is a lack of research on SD in grappling combat sports. Comparing BJJ and overhead sports and strike combat sports may not be appropriate; therefore, more research is necessary.

The prevalence of SD in athletes with shoulder injuries was between 67% and 100%; however, this is most commonly seen in overhead sports [17,35]. BJJ is a martial art that involves gripping, ground grappling, and joint locks [36,37,38]. Judo is similar to BJJ, and shoulder lateral asymmetry (70%) and a winged scapula (56%) were identified in 50 judo athletes [39]. Our BJJ results are similar to these judo results. However, we identified differences between the dominant and non-dominant arms. Over five years of BJJ training and higher levels of BJJ belts (purple, brown, and black belts) showed a higher prevalence of SD in the non-dominant arm (Table 2 and Table 3). Advanced BJJ athletes have reported more shoulder injuries than beginners [3]. Greater exposure to training and competition is related to a greater prevalence of injuries in BJJ athletes [3], which could lead to shoulder weakness and malfunction, causing SD [14,15]. Judo athletes exhibited forward shoulder posture (20%) and forward head posture (58%) along with SD [39]. Boxers showed a higher prevalence of obvious SD in the non-dominant arm (15.27%) than in the dominant arm (6.94%) [13]. Boxers launch more punches with their non-dominant arms [40]. Judo athletes also prefer a specific lateral side for combat [41,42,43]. BJJ athletes are similar to other combat sport athletes, preferring a specific lateral side to compete [41], which could lead to muscular skeletal imbalances, increasing the chance of shoulder injury and SD [15,44,45]. Our results with non-dominant arm SD in BJJ athletes may be related to favoring a specific side during training and combat. Judo is a grappling combat sport, but BJJ and judo are different sports. Boxing is a combat sport, but it is a strike sport. Therefore, judo and boxing may exhibit different results compared to BJJ.

The shoulder internal rotation isometric strength weakened as the severity of SD worsened in the dominant arm (*p* = 0.014). External rotation isometric strength showed a similar trend; however, the difference was not significant (*p* = 0.054). A previous study reported weakness in shoulder rotational strength with SD [46,47]. Judo athletes with a history of shoulder injury show reduced shoulder rotational strength [48]. Elite judo athletes show increased strength in the dominant arm, but the relationship of this with SD has not been reported [49]. An imbalance in shoulder rotational strength may be related to shoulder injury [2,44]. Upper extremity injuries are the most common in all age groups of BJJ athletes [9]. SD could decrease the efficiency of shoulder function, which could be related to the weakness of the rotator cuff [15,34]. The recovery of rotator cuff strength is required to optimize shoulder function [50,51]. Early detection and intervention could improve shoulder function and decrease the risk of shoulder injury [16,52]. Therefore, identifying SD and shoulder rotational strength in BJJ athletes is recommended for the recovery and prevention of shoulder injuries. However, further studies on BJJ athletes with SD are necessary to improve shoulder function and reduce injuries.

There was no significant difference in isometric neck strength between the normal and SD groups. However, neck extension and lateral flexion were weaker in the obvious SD group than in the normal SD group (Table 5). The scapula connects the shoulder and cervical spine and provides mobility and stability to the neck and shoulders [19]. Neck muscles can affect scapular movement because the neck and scapula share muscle attachments [22]. Because of the relationship between the neck and scapula, SD affects cervical motion and stability [19]. However, the periscapular muscle activity in chronic neck pain is not significantly different from that in healthy individuals [21]. There was no significant difference in shoulder strength between healthy individuals with or without SD [27]; however, there is a lack of research on SD and neck strength in BJJ athletes. BJJ athletes show a higher risk of neck injuries [23]. Other studies have reported that the prevalence of neck injury is lower than that of other joint injuries in BJJ athletes [5,11]. More than half of all BJJ injuries are caused by takedowns and submissions [24]. Furthermore, the BJJ triangle technique (choke) can cause neck injuries [2]. Therefore, further research on neck strength, neck injury, and SD is necessary.

Although the obvious SD group showed higher NDI scores compared with the normal SD group, there was no significant difference in the NDI scores with and without SD in the dominant or non-dominant arm. SD is associated with neck pain [22]. SD has been identified in individuals with neck pain and altered periscapular activation [19,22]. The trapezius and serratus anterior muscles are important scapular stabilizers for three-dimensional scapular motion [14,34]. NDI scores were higher in the obvious SD group than in the normal SD group in elite boxing [13]. The prevalence of neck injury has been reported to be higher in non-strike combat sports (19%) than in strike sports (11%) [6]. Boxing is a strike combat sport, and repeated punching can cause microtrauma that can lead to SD [13,15]. BJJ is a non-strike combat sport involving ground techniques that can lead to neck injuries [2,24]. Neck strengthening exercises decrease neck pain and disability in martial art athletes [53]. Scapular-focused exercises decrease neck pain and disability [54]. The BJJ training period and isometric shoulder internal rotation were significantly different between the normal and obvious SD groups. However, neck isometric strength and disability were not significantly different between the two groups. Therefore, SD in BJJ athletes might be related to the training period and shoulder strength rather than neck strength and disability. Moreover, pain threshold and tolerance are higher in combat and contact sports athletes [55,56,57]. There were 26 (54.1%) BJJ athletes with no neck disability, 20 (41%) with a mild disability, and only 2 (4%) with moderate disability. No severe or complete disabilities were reported. Although our results showed no significant difference in neck disability between athletes with and without SD, further studies are necessary to investigate the relationship between neck disability and SD in BJJ athletes.

This study had several limitations. We recruited 48 BJJ athletes from South Korea; however, there were 42 male and 6 female BJJ athletes. Male BJJ athletes showed a significant difference in shoulder internal rotation isometric strength; however, female athletes showed the opposite results. Further studies are necessary to identify sex differences in shoulder strength with and without SD. In addition, the participants were recruited from a limited area, which may have affected the results. Another limitation is the combat style of BJJ. Guard fighters have more upper limb injuries than pass fighters [45]. Different combat styles may affect the shoulders and SD. The third limitation of this study is that there is no gold standard test for SD. Although we applied SDT with a high reliability method (ICC = 0.86, k = 0.57–0.65) [29,58], identifying SD relies on a clinician’s knowledge and experience.

## 5. Conclusions

The prevalence of SD in BJJ athletes was 64.6%, which was as high as that in overhead athletes. The nondominant arm showed a higher prevalence of SD and more severe SD than the dominant arm. Moreover, BJJ athletes with more than five years of training had a higher prevalence of SD in the nondominant arm. Shoulder internal rotation isometric strength was significantly weaker in the nondominant arm of BJJ athletes. Neck strength and disability did not differ significantly between individuals with or without SD.

## Figures and Tables

**Figure 1 medicina-59-01481-f001:**
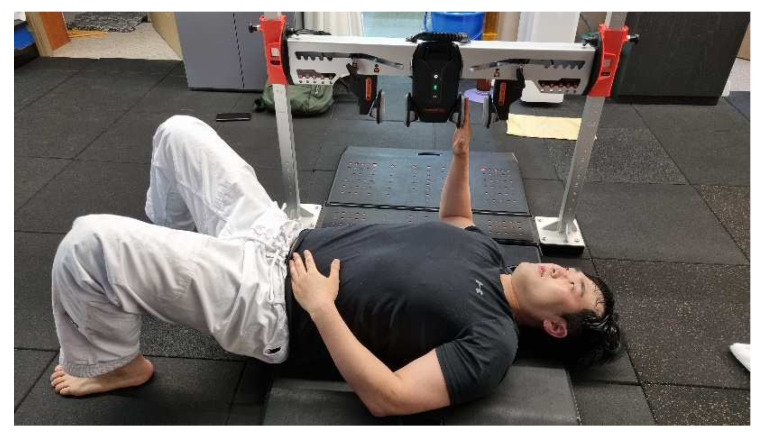
Measuring shoulder isometric strength.

**Figure 2 medicina-59-01481-f002:**
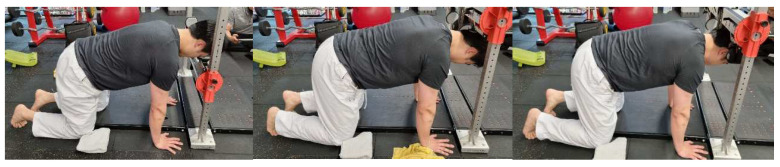
Measuring neck isometric strength.

**Table 1 medicina-59-01481-t001:** The prevalence of scapular dyskinesis in Brazilian ju-jitsu.

Scapular Dyskinesis	Normal	Subtle	Obvious
Dominant arm (*n* = 48)	12 (25%)	18 (37.5%)	18 (37.5%)
Non-dominant arm (*n* = 48)	7 (14.6%)	19 (39.6%)	22 (45.8%)
Total (*n* = 96)	19 (19.8%)	37 (38.5%)	40 (41.7%)

**Table 2 medicina-59-01481-t002:** The BJJ training period with scapular dyskinesis.

Scapular Dyskinesis	Training Period	*n*	Mann–Whitney	Z	*p*
Dominant arm (*n* = 48)	1–5 years	24	273	−0.33	0.742
	Over 5 years	24			
Non-dominant arm (*n* = 48)	1–5 years	24	166	−2.746	0.006 **
	Over 5 years	24			

** *p* < 0.01. BJJ, Brazilian ju-jitsu.

**Table 3 medicina-59-01481-t003:** The BJJ belts with scapular dyskinesis.

Scapular Dyskinesis	Belt (Level)	*n*	Mann–Whitney	Z	*p*
Dominant arm (*n* = 48)	White/blue	21	256.5	−0.598	0.55
	Purple/Brown/blue	27			
Non-dominant arm (*n* = 48)	Whit/blue	21	176.5	−2.428	0.015 *
	Purple/Brown/Black	27			

* *p* < 0.05. BJJ, Brazilian ju-jitsu.

**Table 4 medicina-59-01481-t004:** Dominant arm isometric rotation strength with scapular dyskinesis.

		Normal ^a^ (N)	Subtle ^b^ (N)	Obvious ^c^ (N)	F	*p*	Bonferroni
Dominant arm (*n* = 48)	Shoulder Internal rotation	100.43 ± 24.23	96.43 ± 20.95	78.10 ± 21.80	4.688	0.014 *	a > c (0.028) b > c (0.049)
		(85.03–115.82)	(86.01–106.85	(67.25–88.94)			
	Shoulder External rotation	101.12 ± 37.95	99.34 ± 24.49	77.83 ± 28.92	3.119	0.054	
		(77.01–125.23)	(87.16–111.52)	(63.45–92.21)			
Non-dominant arm (*n* = 48)	Shoulder Internal rotation	89.59 ± 16.85	91.04 ± 30.76	89.80 ± 20.28	0.016	0.984	
		(74.01–105.17)	(76.22–105.87)	(80.81–98.79)			
	Shoulder External rotation	96.75 ± 15.60	92.40 ± 40.56	90.89 ± 24.86	0.093	0.911	
		(82.32–111.18)	(72.85–111.95)	(79.87–101.91)			

* *p* < 0.05.

**Table 5 medicina-59-01481-t005:** Neck isometric strength with scapular dyskinesis.

Isometric Strength (N)		Normal ^a^ (N)	Subtle ^b^ (N)	Obvious ^c^ (N)	F	*p*
Dominant arm (*n* = 48)	Neck Flexion	191.69 ± 52.94	209.50 ± 49.17	196.00 ± 74.68	0.37	0.693
		(158.06–225.33)	(185.05–233.95)	(158.86–233.13)		
	Neck Extension	305.07 ± 84.19	307.24 ± 81.24	274.53 ± 89.96	0.785	0.462
		(251.58–358.56)	(266.84–347.64)	(229.79–319.26)		
	Neck Lateral Flexion	195.28 ± 86.38	202.53 ± 75.70	154.25 ± 68.95	0.771	0.469
		(140.39–250.16)	(164.89–240.18)	(119.96–188.54)		
Non-dominant arm (*n* = 48)	Neck Flexion	208.30 ± 41.41	201.91 ± 70.38	195.67 ± 57.35	0.129	0.88
		(170.00–246.60)	(167.99–235.83)	(170.24–221.10)		
	Neck Extension	308.40 ± 62.76	295.95 ± 97.11	288.67 ± 82.71	0.143	0.867
		(250.36–366.44)	(249.14–342.75)	(252.00–325.34)		
	Neck Lateral Flexion	197.70 ± 39.33	169.02 ± 88.61	169.97 ± 61.57	1.95	0.154
		(161.33–234.07)	(126.31–211.73)	(142.67–197.27)		

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
