# Peer review of "Association of Scapular Dyskinesis with Neck and Shoulder Function and Training Period in Brazilian Ju-Jitsu Athletes"

_medicina, 2023, doi:10.3390/medicina59081481_

Round 1
Reviewer 1 Report
Dear authors,
Thank you very much for your efforts in your research. The subject of your research is valuable in terms of its scope and content, but some shortcomings are clearly visible. After these deficiencies are corrected, I find your article suitable for publication in the journal Medicina.
Required revision requests are below.
Introduction
-Please add a short sentence about BJJ at the beginning of the introduction. BJJ football is not as popular as basketball, so readers can learn about the branch.
-Please write the main hypothesis of your research after the purpose statement.
method
Please 2.1. Add the article you have calculated G power as a reference in the participants section.
There is no need for any revisions in the Results and Discussion sections.
Author Response
Reviewer 1
Dear authors,
Thank you very much for your efforts in your research. The subject of your research is valuable in terms of its scope and content, but some shortcomings are clearly visible. After these deficiencies are corrected, I find your article suitable for publication in the journal Medicina.
Thank you, so much for your review. We have attached the corrections following your comments below.
Required revision requests are below.
Introduction
-Please add a short sentence about BJJ at the beginning of the introduction. BJJ football is not as popular as basketball, so readers can learn about the branch.
Brazilian ju-jitsu (BJJ) is a grappling martial arts emphasizing takedowns, joint locks, and chokeholds to win with different fight styles [1-3]. Although BJJ athletes are not exposed to direct blows and kicks, the prevalence of injuries such as fractures, dislocations, sprains, muscle strains, and tendinopathy is very high [3, 4, 5].
-Please write the main hypothesis of your research after the purpose statement.
Therefore, this study aimed to investigate SD in BJJ athletes with neck and shoulder pain and disability and the relationship of SD with strength and duration of BJJ training. Our hypothesis was BJJ athletes with SD would present more neck and shoulder pain and disability with weaker neck and shoulder strength.
method
Please 2.1. Add the article you have calculated G power as a reference in the participants section.
The sample size was calculated using G Power software (University of Kiel, Kiel, Germa-ny). With an effect size of 0.8 from the previous study [54], a significance level of 0.05, and a power of 0.95, we required a sample size of 30.
There is no need for any revisions in the Results and Discussion sections.

Reviewer 2 Report
Thank you for this opportunity to review this article, I hope it will be useful and that my comments will improve your work.
Introduction:
Regarding the acronyms you use in the text: what does AC stand for?
I recommend you to rethink the introduction, organise the information, from anatomical region to its correlation with other sports.
Review this sentence, it does not provide information of interest, according to the approach of the study: "However, sweeping and tumbling 34 can lead to scoring and winning matches" (34-35).
Do you get conclusions in your introduction? I recommend that you rethink it.
"A high rate of neck and shoulder injuries in BJJ athletes could lead to SD, which is associated with neck pain and shoulder injuries." (39-40)
Discussion
What relationship do you find in your study when comparing Ju-Jitsu with baseball or boxing? There is no relationship between the variables they have analysed.
"Baseball pitchers exhibited differences in shoulder rotation strength with SD [40]. Boxers showed a higher prevalence of obvious SD in the non-dominant arm (15.27%) than in the dominant arm (6.94%) [13]. Boxers launched more punches with their non-dominant arms [41] " (210-212)
These sentences do not fit in the section.
"The rate of BJJ injury is increasing [23]; however, there is a lack of post-injury research on BJJ [24]. There- fore, identifying SD and shoulder strength in BJJ athletes is recommended for the recovery and prevention of shoulder injuries. However, further studies on BJJ athletes with SD are necessary to improve shoulder function and reduce injuries." (230-234)
"further studies are necessary to investigate".
You use similar expressions in different paragraphs. They can remove them and refer to them in general in conclusions.
You should continue their line of research, in general their study only provides an incidence of shoulder injuries similar to previous studies, it would be of interest to focus their research on the risk movements in the practice of Ju-jitsu, which can cause shoulder and neck injuries.
Author Response
Reviewer 2
Thank you for this opportunity to review this article, I hope it will be useful and that my comments will improve your work.
Thank you, so much for your review. We have attached the corrections following your comments below.
Introduction:
Regarding the acronyms you use in the text: what does AC stand for?
Seventy-three percent of acromioclavicular (AC) joint separations showed scapular dys-kinesis (SD), which is a malfunction of the scapula
I recommend you to rethink the introduction, organise the information, from anatomical region to its correlation with other sports.
Our topic goes from introduction of BJJ and neck shoulder injuries, SD and prevalence in sports, SD and anatomical background with sports, then purpose of this study why SD study is necessary in BJJ. We tried to focus on sports related SD, however, BJJ is grappling combat sports, and there is no research in English investigated on grappling combat sports. We could not find any research directly related on. Therefore, the background might be weak, however, we found high prevalence of SD and related background with BJJ. We hope this research would initiate SD research in grappling combat sports.
Review this sentence, it does not provide information of interest, according to the approach of the study: "However, sweeping and tumbling 34 can lead to scoring and winning matches" (34-35).
Seventy-five percent of shoulder injuries in BJJ athletes occur during sweeping and tum-bling [9]. However, sweeping and tumbling can lead to scoring and winning matches [10-12]. Therefore, BJJ athletes are willing to use those techniques, and they are at a risk of sustaining shoulder injuries [9].
We changed the following sentence to support the weak sentence.
Do you get conclusions in your introduction? I recommend that you rethink it.
"A high rate of neck and shoulder injuries in BJJ athletes could lead to SD, which is associated with neck pain and shoulder injuries." (39-40)
A high rate of neck and shoulder injuries in BJJ athletes might be related to SD, which is associated with neck pain and shoulder injuries.
Discussion
What relationship do you find in your study when comparing Ju-Jitsu with baseball or boxing? There is no relationship between the variables they have analysed.
There is no SD research on grappling sports, but there is a few on strike combat sports such as boxing. Because of the lack, we could not find the right research to cite. Therefore, we added the sentences at the end of the first paragraph of the discussion
However, there is a lack of research on SD in grappling combat sports. Comparing BJJ and overhead sports and strike combat sports may not appropriate, therefore, more research is necessary.
"Baseball pitchers exhibited differences in shoulder rotation strength with SD [40]. Boxers showed a higher prevalence of obvious SD in the non-dominant arm (15.27%) than in the dominant arm (6.94%) [13]. Boxers launched more punches with their non-dominant arms [41] " (210-212)
Greater exposure to training and competition is related to a greater prevalence of injuries in BJJ athletes [3], which could lead to shoulder weakness and malfunction, causing SD [15,16]. Judo athletes exhibited forward shoulder posture (20%) and forward head posture (58%) along with SD [39] . Boxers showed a higher prevalence of obvious SD in the non-dominant arm (15.27%) than in the dominant arm (6.94%) [14]. Boxers launch more punches with their non-dominant arms [41]. Judo athletes also prefer a specific lateral side for combat [42-44]. BJJ athletes are similar to other combat sport ath-letes, preferring a specific lateral side to compete [42], which could lead to muscular skel-etal imbalances, increasing the chance of shoulder injury and SD [43, 16, 46]. Our results with non-dominant arm SD in BJJ athletes may be related to favoring a specific side during training and combat. Judo is grappling combat sports, but BJJ and judo are different sports. Boxing is combat sport, but it is strike sport. Therefore, judo and boxing may exhibit different results compared to BJJ.
These sentences do not fit in the section.
"The rate of BJJ injury is increasing [23]; however, there is a lack of post-injury research on BJJ [24]. There- fore, identifying SD and shoulder strength in BJJ athletes is recommended for the recovery and prevention of shoulder injuries. However, further studies on BJJ athletes with SD are necessary to improve shoulder function and reduce injuries." (230-234)
We deleted “the rate of BJJ injury….” To clearly present the direction from the result of our study only.
Early detection and intervention could improve shoulder function and decrease the risk of shoulder injury [17,53]. Therefore, identifying SD and shoulder rotational strength in BJJ athletes is recommended for the recovery and prevention of shoulder injuries. However, further studies on BJJ athletes with SD are nec-essary to improve shoulder function and reduce injuries.
"further studies are necessary to investigate".
You use similar expressions in different paragraphs. They can remove them and refer to them in general in conclusions.
We deleted the sentence from the conclusions
The prevalence of SD in BJJ athletes was 64.6%, which was as high as that in over-head athletes. The nondominant arm showed a higher prevalence of SD and more severe SD than the dominant arm. Moreover, BJJ athletes with more than 5 years of training had a higher prevalence of SD in the nondominant arm. Shoulder internal rotation isometric strength was significantly weaker in the nondominant arm of BJJ athletes. Neck strength and disability did not differ significantly between individuals with or without SD.
You should continue their line of research, in general their study only provides an incidence of shoulder injuries similar to previous studies, it would be of interest to focus their research on the risk movements in the practice of Ju-jitsu, which can cause shoulder and neck injuries.
Thank you for your direction of research. Although there is few research about scapular dyskinesis in combat sports, I see tremendous amount of combat sports athletes with SD, and they have shoulder problems. I will continue to investigate on those athletes with their neck and shoulder.

Round 2
Reviewer 2 Report
Your article has improved with your corrections. I recommend that you continue your line of research.